# OpenReview forum: "Sherkala-Chat: Building a State-of-the-Art LLM for Kazakh in a Moderately Resourced Setting"
_colmweb.org/COLM/2025/Conference — COLM 2025_

### Official Review · Reviewer_EVms · 2025-05-04

**Rating:** 6
**Confidence:** 4
**Ethics Flag:** 1

**Summary:**

This paper introduces Sherkala-Chat (8B), an instruction-tuned open-weight large language model specifically designed for Kazakh, a moderately resourced language. The authors adapt LLaMA-3.1-8B through continuous pre-training on 45.3B tokens (including 19.45B Kazakh tokens) and employ a custom tokenizer that reduces token usage for Kazakh by 56.8%. The model undergoes instruction tuning with multilingual datasets and Kazakhstan-specific content. Comprehensive evaluations across knowledge, reasoning, text generation, and safety benchmarks demonstrate that Sherkala-Chat outperforms existing open multilingual and Kazakh-specific models while maintaining competitive performance in English and Russian.

**Questions To Authors:**

1. While the overall data ratio is 3:1:3 (Kazakh:Russian+Turkish:English), does the paper address any special handling of batch-level data composition during training?

2.  What specific benefits were observed from the chosen embedding initialization approach, and how were these measured?

3. What setup differences existed between the base model (Sherkala) and chat model (Sherkala-Chat) during evaluation tasks?

4. How might the reliance on GPT-4o as a judge for generation quality impact the assessment results? Could this introduce biases in the evaluation process?

**Reasons To Accept:**

1. Addresses an important gap in AI accessibility for underrepresented languages, contributing to linguistic diversity in NLP.
2. Clear methodological contributions including a custom tokenizer design and embedding initialization approach specifically optimized for Kazakh.
3. Strong empirical results with significant performance improvements over existing models (+3.9 points over KazLLM-1.0, +9.9 points over Irbis-v0.1).
4. Comprehensive evaluation framework covering multiple dimensions (downstream tasks, generation, safety) and multiple languages.
5. Well-documented training process that enhances reproducibility and supports further research in low-resource language adaptation.
Practical impact through open-weight release that enables real-world applications for Kazakh speakers.

**Reasons To Reject:**

1. **Reliance on machine-translated Kazakh content**,the dataset includes synthetic data generated by translating English Wikipedia articles into Kazakh using Google Translate, accounting for 24% of the Kazakh corpus. Without rigorous validation of translation quality, this approach may introduce artificial patterns that don't reflect authentic Kazakh language usage, potentially limiting the model's cultural relevance.

2. **Lack of qualitative examples for Kazakh-specific tasks**, while the paper presents extensive quantitative benchmarks, it doesn't provide concrete examples of model responses to Kazakh-specific queries that would demonstrate its ability to handle culturally nuanced content, historical references, or Kazakhstan-specific knowledge - areas where improvement over general multilingual models would be most valuable.

---

> ### Author Response · Authors · 2025-06-03
> **Response to reviewer EVms**
>
> Thank you for your positive feedback on our paper. Please find our feedbacks on your comments:
>
> > Reliance on machine-translated Kazakh content
>
> We would like to clarify that 76% of the Kazakh tokens used in pretraining are not translation-based (as noted in line 93). While we do rely on translation in some stages—primarily due to the limited availability of high-quality Kazakh datasets—we have made substantial efforts to include authentic Kazakh data. Specifically, we curated original Kazakh datasets for instruction fine-tuning and evaluation tasks, including multiple-choice questions, generation, and safety benchmarks. These components ensure that our evaluation is grounded in high-quality, native Kazakh content and not solely dependent on translation.
>
> Regarding translation quality, we conducted a preliminary study to assess the quality of the translated training data. Specifically, we compared the performance of GPT-4o, Google Translate, and the open-source SeamlessM4T system on Kazakh translation using the FLORES benchmark. Based on BLEU scores, Google Translate achieved the highest score at 40.6 for EN2KAZ and 25.6 for EN2KAZ, indicating better translation quality among the evaluated systems.
>
> > Lack of qualitative examples for Kazakh-specific tasks
>
> Our evaluation includes KazMMLU, a non-translation-based dataset developed around the local education system in Kazakhstan. For generation tasks, we use two in-house datasets—CultSet and GovSet—which focus on culturally and government-relevant topics specific to Kazakhstan. Additionally, our safety evaluation includes culturally grounded scenarios, as described in Section 4. Across these Kazakh-specific evaluation datasets, Sherkala consistently outperforms baseline models. To make this more concrete, we will include representative prompt–response examples from CultSet and GovSet in the Appendix in the next revision.
>
>
> > While the overall data ratio is 3:1:3 (Kazakh:Russian+Turkish:English), does the paper address any special handling of batch-level data composition during training?
>
> We use a simple mixed-language setup during pretraining. However, to strengthen Kazakh representation, we upsample the Kazakh dataset by 20% at the end of each epoch. We will clarify this detail in the next revision.
>
> > What specific benefits were observed from the chosen embedding initialization approach, and how were these measured?
>
> We follow the embedding initialization approach proposed by Gosal et al. (ICML 2024), which has been shown to be effective in similar settings. We believe relying on this state-of-the-art method provides a solid foundation for the adaptation process.
>
> > What setup differences existed between the base model (Sherkala) and chat model (Sherkala-Chat) during evaluation tasks?
>
> There is no significant difference in the evaluation setup between the base and chat models. The only distinction is that the chat model uses a chat template to align with instruction fine-tuning. This template follows the same format used by LLaMA-3.1.
>
> > How might the reliance on GPT-4o as a judge for generation quality impact the assessment results? Could this introduce biases in the evaluation process?
>
> In addition to using GPT-4o for evaluation, we also conducted manual assessments to mitigate potential bias. As noted on lines 1172 and 1189, GPT-4o achieved an accuracy of 90.4% in the safety evaluation. For the generation evaluation on GovSet and WikiSet (Figure 2), human evaluation results are reported in Laiyk et al. (ACL 2025). This combination of automatic and human evaluation helps ensure the reliability of our assessment.

---

> > ### Comment · Reviewer_EVms · 2025-06-08
> >
> > Thank you for the detailed response addressing the raised concerns. Given the current abundance of work on language expansion for large language models, I maintain the original rating of 6 after comprehensive consideration.

---

### Official Review · Reviewer_b2Hx · 2025-05-07

**Rating:** 6
**Confidence:** 3
**Ethics Flag:** 1

**Summary:**

In this paper, the authors have demonstrated the full pipeline for training an LLM for Kazakh, covering the core steps including continual pre-training, instruction tuning, and safety alignment. The experimental results show significant improvements on Kazakh benchmarks compared with the base model and another counterpart.

**Questions To Authors:**

1. Do you have the plan to release your training dataset in Kazakh?
2. Is there any human evaluation about the quality for both the translated training set and evaluation set?

**Reasons To Accept:**

1. The technical report keeps the full pipeline transparent to the community, which can be beneficial to other low resource language community.
2. The experimental results are solid.

**Reasons To Reject:**

1. Though it may not be necessary, compared with the conventional training pipeline, it seems that the method in this paper does not bring some new insights to the community as how to fast adapt an existing LLM to a new **low resource language**.
2. The usage of machine-translated dataset raises some concern about the data quality: if the SOTA models, e.g., ChatGPT and Google Translate (Gemini) can already do pretty well on Kazakh, what's the main value of this work to the Kazakh community?
3. There are only two solid benchmarks for Kazakh (KazMMLU) and MIS Math, which can also raise some concern whether the released model can help complete real world tasks based on Kazakh. Besides, using GPT-4o as evaluator to judge the generation performance on Vicuna-Instructions-80 and MT-Instructions-80 is also not reliable, as I'm not sure 4o's capability on Kazakh.

---

> ### Author Response · Authors · 2025-06-03
> **Response to reviewer b2Hx**
>
> Thank you for your positive assessment of our work. Please find our detailed responses below:
>
> > compared with the conventional training pipeline, it seems that the method in this paper does not bring some new insights to the community as how to fast adapt an existing LLM to a new low resource language.
>
> The goal of this paper is to show how a multilingual LLM can be effectively adapted into a Kazakh-centric model, while addressing the practical challenges of limited resources for pretraining, instruction-tuning, and evaluation. Our contributions include insights into language mixture strategies for efficient adaptation, as well as the development of in-house datasets for cultural instruction tuning and safety alignment—resources that did not previously exist for Kazakh.
>
> In addition, our evaluation is extensive, covering multiple aspects such as knowledge, reasoning, misinformation, bias, generation quality, safety, and cultural alignment. These evaluations span across Kazakh, Russian, and English, offering a broad and rigorous assessment framework. We believe this comprehensive approach provides valuable insights into the adaptation of LLMs for low-resource and culturally specific contexts.
>
>
> > The usage of machine-translated dataset raises some concern about the data quality: if the SOTA models, e.g., ChatGPT and Google Translate (Gemini) can already do pretty well on Kazakh, what's the main value of this work to the Kazakh community?
>
> While ChatGPT and Gemini perform well in Kazakh, they are commercial systems and not directly comparable to our fully open-source model. Sherkala is designed to be openly accessible and can be adopted by public institutions and government agencies in Kazakhstan, something that is not feasible with proprietary models. More importantly, Sherkala contributes to the research community by providing an open, reproducible foundation for studying language modeling, instruction tuning, and cultural alignment in low-resource settings, an area not addressed by commercial models.
> In terms of performance, Sherkala outperforms other open-source models with over 70B parameters on Kazakh benchmarks, as demonstrated in the Kaz-Offline Arena.
> Regarding data quality, our evaluation is based on a mix of translated and original Kazakh and Russian datasets, including KazMMLU, NisMATH, USE, CultSet, GovSet, and a safety evaluation set, all of which are high-quality and human-verified. This evaluation setup is a core contribution of our work, offering a comprehensive assessment across multiple tasks and three languages, as detailed in our previous response.
> > There are only two solid benchmarks for Kazakh (KazMMLU) and MIS Math, which can also raise some concern whether the released model can help complete real world tasks based on Kazakh. Besides, using GPT-4o as evaluator to judge the generation performance on Vicuna-Instructions-80 and MT-Instructions-80 is also not reliable, as I'm not sure 4o's capability on Kazakh.
>
> To clarify, in addition to KazMMLU and NisMATH, we also include high-quality evaluation datasets for safety and generation: specifically, CultSet and GovSet. These go beyond general-purpose benchmarks and are designed to reflect real-world and culturally grounded use cases. For the Safety evaluation, we conducted manual assessment, achieving 90% accuracy (see line 1170). For CultSet and GovSet, evaluation details and evidence are documented in the paper we cite (Laiyk et al., 2025). While we used GPT-4o for Vicuna-Instructions-80 and MT-Instructions-80, these are complemented by our own domain-specific evaluations to provide a more complete and reliable assessment.
>
> > Do you have the plan to release your training dataset in Kazakh?
>
> We are currently reviewing the redistribution rights for the Kazakh pretraining data. While we strongly support open-source principles, we need to ensure full compliance with licensing terms before making it publicly available.
>
> As for our safety dataset and in-house instruction fine-tuning datasets—CultSet and GovSet—we plan to release them under a permissive open-source license to support further research and development in the community.
>
> > Is there any human evaluation about the quality for both the translated training set and evaluation set?
>
> We conducted a preliminary study to assess the quality of the translated training data. Specifically, we compared the performance of GPT-4o, Google Translate, and the open-source SeamlessM4T system on Kazakh translation using the FLORES benchmark. Based on BLEU scores, Google Translate achieved the highest score at 40.6 for EN2KAZ and 25.6 for EN2KAZ, indicating better translation quality among the evaluated systems.

---

> > ### Comment · Reviewer_b2Hx · 2025-06-09
> >
> > Thanks for the authors' efforts for addressing my concern. I would keep my original score.

---

### Official Review · Reviewer_uizC · 2025-05-13

**Rating:** 5
**Confidence:** 4
**Ethics Flag:** 1

**Summary:**

This paper introduces Sherkala-Chat, a 8B LLM built upon LLama-3.1, which is enhanced to support Kazakh speakers. Specifically, it experiences continual pretraining with 45.3B tokens across Kazakh, English, Russian, and Turkish. With 8 billion parameters, it seems that this LLM outperforms existing open Kazakh and multilingual models of similar scale while achieving competitive performance in English. For alignment, they leverage *translated* instruction datasets, a Kazakhstan-specific instruction dataset that is automatically constructed and manually verified, and Kazakh-specific safety data. They finally open-source their model in the huggingface.

**Questions To Authors:**

N/A

**Reasons To Accept:**

* It's good to see work that emphasizes extending LLMs to low-resource languages. This paper focuses on Kazakh.
* I am happy to see that they have open-sourced their LLM in the Huggingface, although it could be better if they open-source their training data as well, which I believe could be beneficial to the field.
* Extensive evaluations show that their model demonstrates near state-of-the-art performance among models with similar sizes.

**Reasons To Reject:**

* I feel like one noticeable limitation of this paper is their use of translated data for instruction tuning (Table 2). Though with another in-house dataset curated with human evaluation, I am not sure how much space such data occupies.. Let me know if I am wrong. I make such inference from Table 2. Thus, I doubt about the utility of their LLM in terms of culture aspects, especially regarding evaluations around linguistic nuances. A small safety evaluation might not be enough (line 1147).
* I am not convinced enough regarding "Sherkala-Chat achieves competitive performance in English". Instead, in Table 8,  Sherkala-Chat is worsen than Llama-3.1-8B-Instruct in English.
* I acknowledge that Sherkala-Chat is promising in Kazakh against models with similar sizes. However, in the comparisons, there are some updates regarding the compared models as well, such as Qwen, Deepseek, etc. I am not sure how Sherkala-Chat is compared to them.
* Another criticism of Sherkala-Chat is regarding its utility, while I think a contribution towards low-resource language is definitely meaningful. For example, from Huggingface, the number of downloads is small.

---

> ### Author Response · Authors · 2025-06-03
> **Response to reviewer uizC**
>
> > I feel like one noticeable limitation of this paper is their use of translated data for instruction tuning (Table 2).  I doubt about the utility of their LLM in terms of culture aspects, especially regarding evaluations around linguistic nuances. A small safety evaluation might not be enough
>
> Since there is no publicly available Kazakh instruction-tuning dataset, translation was the most practical approach to initiate this work. In addition, we developed in-house datasets for cultural fine-tuning and safety alignment. Although these datasets are smaller than standard instruction-tuning corpora (as shown in Table 2), they are human-verified and of high quality. To increase their impact, we triple the size of the safety dataset and applied the cultural alignment dataset in the final fine-tuning stage. We will clarify these details in the next revision. In our evaluation, we demonstrate the effectiveness of this approach: our models outperform existing Kazakh-specific LLMs in both safety and culturally grounded generation tasks, as shown on CultSet and GovSet.
>
> > I am not convinced enough regarding "Sherkala-Chat achieves competitive performance in English". Instead, in Table 8, Sherkala-Chat is worsen than Llama-3.1-8B-Instruct in English.
>
> To clarify, Table 8 shows that Sherkala-Chat achieves competitive performance compared to its Kazakh-specific counterpart, KazLLM-1.0-8B. For instance, Sherkala-Chat scores 6.55 on Vicuna evaluation, while KazLLM scores 6.66. In English MCQ evaluation, Sherkala-Chat achieves an average accuracy of 59.1, slightly outperforming KazLLM’s 58.6. However, our primary focus is not English, but Kazakh and Russian—the two major languages spoken in Kazakhstan. In these languages, Sherkala-Chat consistently outperforms KazLLM across multiple evaluation dimensions (see Table 8 and Figure 4).
>
> > I acknowledge that Sherkala-Chat is promising in Kazakh against models with similar sizes. However, in the comparisons, there are some updates regarding the compared models as well, such as Qwen, Deepseek
>
> Thank you for acknowledging the contribution of our work. We agree that model development is rapidly evolving, and new versions of Qwen and Deepseek continue to emerge. However, our primary goal is not to outperform every new general-purpose model, but to demonstrate how LLaMA-3.1 can be effectively adapted into a strong, culturally and linguistically specialized Kazakh model. Sherkala-Chat is the first model of its kind built on LLaMA-3.1 and extensively evaluated across multiple dimensions—including safety, instruction-following, and cultural alignment—in Kazakh and Russian. To our knowledge, no prior work has conducted such a comprehensive evaluation for Kazakh LLMs. We believe this is a valuable step forward for low-resource and culturally grounded language modeling.
>
> > Another criticism of Sherkala-Chat is regarding its utility, while I think a contribution towards low-resource language is definitely meaningful. For example, from Huggingface, the number of downloads is small.
>
> Thank you again for recognizing the significance of our work. Regarding the number of downloads on Hugging Face, we are seeing a steady increase, which we find encouraging. Notably, our model also ranks among the top performers in the Kaz-Offline Arena, even outperforming some 70B-scale models.

---

> ### Comment · Reviewer_uizC · 2025-06-05
> **Official Comment from Reviewer uizC**
>
> Overall, I thank authors for the rebuttal.
>
> > Cultural aspects
>
> Thanks for pointing out the evaluation on CultSet and GovSet. This is also incorporated in your instruction tuning set (perhaps have done the careful separation for evaluation). Sure, it shows some reliability. However, just like the lack of benchmarks for evaluating cultural aspects in Kazakh, your evaluation is not fully convincing—especially considering that CultSet and GovSet do not appear to have been released yet, and the technical report by the same authors only came out this year on arXiv.
>
> > English performance
>
> I agree. I know that your focus is Kazakh. However, please take a further look at your abstract. There seems to be a potential overclaim. As you said "significantly outperforming existing open Kazakh and multilingual models of similar scale while achieving competitive performance in English". So there are two questions raised.
>
> 1. Have you compared your model with currently popular and up-to-date open-source Kazakh or multilingual LLMs?
>
> If not, for example, if you just show that your model outperforms some early models, it can be overclaimed.
>
> 2. competitive performance in English: this means that your model enhances Kazakh without compromising English ability and even strengthening it. The truth is your model is not competitive with other multilingual LLMs, right? And you slightly weakens Llama in processing English, right?
>
> Let me know if I am wrong.
>
> > Other models
>
> Please see my response to the last point I raised. I personally feel it is definitely helpful to compare these models with your model to strengthen the claims you made in the paper. And please let me know why not if your model is really that promising in Kazakh? For instance, Qwen-3 has put their efforts in multilingual enhancement as far as I know. There are models in similar sizes. Please compare, and argue why not. There is a chance to increase the impact of your work. Like what you said in Kaz-Offline Arena, could you show more comparisons in the rebuttal and your paper?
>
> > Utility
>
> Like I said, I think your work, if done properly, has impacts. Yet, current impacts in the community are relatively lower compared to other multilingual LLMs (not those quite popular LLMs, e.g., Qwen, but some LLMs tailored to particular regions). I understand that this may be attributed to the small number of people using this language. Anyway, this slightly weakens the utility of this work, though still can be meaningful if done properly. This criticism is not critical, yet the impact of being published in this conference can be significantly improved if it is broadly discussed in the overall community. I believe that this should be true.
>
>
> I am happy to discuss more, and please let me know more about your work. Thanks!

---

### Official Review · Reviewer_EowQ · 2025-05-17

**Rating:** 8
**Confidence:** 4
**Ethics Flag:** 1

**Summary:**

This paper introduces two Kazakh language models, Sherkala (8B) and its instruction-tuned variant Sherkala-Chat (8B). Along the paper the authors describe the development process for this model, the data mixture selection, the tweaks for the tokenizer, the continual pre-training process, the instruction tuning and the evaluation.

Sherkala (8B) and Sherkala-Chat (8B) are based on the the LLaMA-3.1 model. The pre-training data includes data for English, Russian, Turkish and Kazakh. The authors translate instructions as well as evaluations from English. The authors evaluate their model in all the standard benchmarks in English, Russian and Kazakh. They also perform a generation evaluation and safety evaluations. The authors also develop some novel datasets for Kazakh to better capture local sensitivities.

**Questions To Authors:**

1. Did you do any experiments to show that the fact that benchmarks are translated can inflate the scores for your model (which is also trained in English)?
2. Would you be open to change the license?
3. Given how much you rely on automatic translations, discussions of the validations that you did, such as the one in Appendix G (1143-1150) should probably be better placed in the main body of the paper.
4. Will you release the data as well as all the code along with the models? If yes, what is going to be the license?

**Reasons To Accept:**

1. The paper is well written and clear. It is easy to read and include easy to understand explanations.
2. The authors release models for Kazakh, an underrepresented language, allowing this linguistic community to access some of the technologies that other languages have had for years.
3. The authors perform extensive evaluation, including standard benchmarking, generation evaluation and safety evaluation.
4. The authors manage to show that the inclusion of Kazakh data do not significantly degrade scores in English.
5. The author curate and create novel datasets in oder to account for local sensitivities of the Kazakh linguistic community.
6. The authors commit to release the models
7. The paper has extensive detail about the pre-processing pipeline and all the training details and evaluation results in the appendices.
8. The authors perform interesting ablations to justify their data mixture

**Reasons To Reject:**

1. The authors heavily rely on translation, which is understandable, but there is little work in this paper showing that these translations are not biasing the evaluation, specially in the case of translated benchmarks and translated knowledge.
2. The models will be released under CC-BY-NC-SA-4.0 license which is non comercial and makes derivatives be released under the same license. Moreover, the authors claim that is ok to use their models in a commercial setting which is contradictory to the license choice.

---

> ### Author Response · Authors · 2025-06-03
> **Response to reviewer EowQ**
>
> We thank the reviewer for positive comments on our paper. Please find below our feedback on the weaknesses and questions:
>
> > The authors heavily rely on translation. There is little work in this paper showing that these translations are not biasing the evaluation, specially in the case of translated benchmarks and translated knowledge.
>
> We would like to clarify that 76% of the Kazakh tokens used in pretraining are not translation-based (as noted in line 93). While we do rely on translation in some stages—primarily due to the limited availability of high-quality Kazakh datasets—we have made substantial efforts to include authentic Kazakh data. Specifically, we curated original Kazakh datasets for instruction fine-tuning and evaluation tasks, including multiple-choice questions, generation, and safety benchmarks. These components ensure that our evaluation is grounded in high-quality, native Kazakh content and not solely dependent on translation.
>
> > The models will be released under CC-BY-NC-SA-4.0 license which is non comercial and makes derivatives be released under the same license. Would you be open to change the license?
>
> We appreciate the reviewer’s comment regarding the license. The models were initially released under the CC-BY-NC-SA-4.0 license during their early development phase. However, we will carefully re-evaluate the licensing terms in the next revision.
>
>
> > Did you do any experiments to show that the fact that benchmarks are translated can inflate the scores for your model (which is also trained in English)?
>
> Yes, our preliminary data mixture study included translated benchmarks, as they were the only resources available at the early stage of our research. We observed that incorporating English datasets boosted model performance (see Figure 1). However, it's challenging to isolate whether the improvement comes from the language or the content, as the translated benchmarks cover general domains such as STEM, which may benefit from English pretraining. To ensure a more robust evaluation, we also include original Kazakh datasets: KazMMLU and NisMATH for multiple-choice evaluation, and in-house datasets for generation and safety evaluations.
>
> > Appendix G (1143-1150) should probably be better placed in the main body of the paper.
>
> We agree with the comment. Due to space constraints in the initial submission, we included these details in the Appendix. We will revise accordingly in the camera-ready version.
>
> > Will you release the data as well as all the code along with the models? If yes, what is going to be the license?
>
> We will release the evaluation code without restriction. The evaluation datasets we used—KazMMLU, the safety dataset, and our in-house instruction fine-tuning dataset—will also be made publicly available under a permissive open-source license.
> As for the pretraining data, we are currently reviewing the redistribution rights. While we strongly support open-source principles, we must ensure proper compliance with licensing terms before releasing it.

---

> > ### Comment · Reviewer_EowQ · 2025-06-07
> >
> > I thank the authors for the response.
> >
> > Regarding the translations, I still think a more in-depth study of how this affects the model should be done at some point. Also the license question should probably be addressed for this model and this work to be more broadly usable.
> >
> > I keep my scores as is.

---

### Decision · Program_Chairs · 2025-07-08

**Decision:**

Accept

**Comment:**

This paper presents Sherkala-Chat, an 8B instruction-tuned open LLM designed to support the Kazakh language through multilingual pretraining and culturally-aligned fine-tuning. The paper contributes a valuable resource for an underrepresented language, supported by comprehensive evaluations and open-weight release. Strengths include the thoughtful inclusion of Kazakh-specific safety and cultural datasets, detailed training pipeline, and clear writing. However, concerns remain about the heavy reliance on translated datasets (as raised by many reviewers) and limited direct comparison with newer multilingual models. Overall it's a meaningful contribution to low-resource NLP, but I would suggest the authors to at least include more discussions on the translated data in the camera-ready version.